# Usage of Cell-Free Protein Synthesis in Post-Translational Modification of μ-Conopeptide PIIIA

**DOI:** 10.3390/md21080421

**Published:** 2023-07-25

**Authors:** Yanli Liu, Zitong Zhao, Yunyang Song, Yifeng Yin, Fanghui Wu, Hui Jiang

**Affiliations:** State Key Laboratory of NBC Protection for Civilian, Beijing 102205, China

**Keywords:** conopeptide, post-translational modification, hydroxylation, cell-free protein system

## Abstract

The post-translational modifications of conopeptides are the most complicated modifications to date and are well-known and closely related to the activity of conopeptides. The hydroxylation of proline in conopeptides affects folding, structure, and biological activity, and prolyl 4 hydroxylase has been characterized in *Conus literatus*. However, the hydroxylation machinery of proline in conopeptides is still unclear. In order to address the hydroxylation mechanism of proline in μ-PIIIA, three recombinant plasmids encoding different hybrid precursors of μ-PIIIA were constructed and crossly combined with protein disulfide isomerase, prolyl 4 hydroxylase, and glutaminyl cyclase in a continuous exchange cell-free protein system. The findings showed that prolyl 4 hydroxylase might recognize the propeptide of μ-PIIIA to achieve the hydroxylation of proline, while the cyclization of glutamate was also formed. Additionally, in *Escherichia coli,* the co-expression plasmid encoding prolyl 4 hydroxylase and the precursor of μ-PIIIA containing pro and mature regions were used to validate the continuous exchange cell-free protein system. Surprisingly, in addition to the two hydroxyproline residues and one pyroglutamyl residue, three disulfide bridges were formed using Trx as a fusion tag, and the yield of the fusion peptide was approximately 20 mg/L. The results of electrophysiology analysis indicated that the recombinant μ-PIIIA without C-terminal amidate inhibited the current of hNa_V_1.4 with a 939 nM IC_50_. Our work solved the issue that it was challenging to quickly generate post-translationally modified conopeptides in vitro. This is the first study to demonstrate that prolyl 4 hydroxylase catalyzes the proline hydroxylation through recognition in the propeptide of μ-PIIIA, and it will provide a new way for synthesizing multi-modified conopeptides with pharmacological activity.

## 1. Introduction

Conopeptides are a kind of small bioactive peptide that is ribosomally synthesized and has multiple disulfides and a large number of post-translational modifications (PTMs). They are characterized by their structural stability, target specificity, and relatively small size and are regarded as a rich source of molecular probes in neuroscience. It is estimated that approximately 50,000 conopeptides could be secreted by different Conus species, and over 10,000 conopeptide sequences have been published [1]. However, only a small number of conopeptides have been used as tools for drug development and in pharmacological research because there are few resources and many complex PTMs that are difficult to achieve in vitro using chemical synthesis and biosynthesis techniques. 

The PTMs of conopeptides are the most complex modifications in marine natural products and are closely related to the structure and bioactivity of conopeptides [2]. For instance, carboxylation of glutamic acid cannot improve the oxidative folding efficiency of tx9a in the presence of calcium [3], but it can stabilize the structure by inducing the formation of α-helix in conantokin [4]. The proper formation of disulfide bridges is essential for bioactivity [5]. The maturation of bioactive conopeptides is produced by proteolytic cleavage of the precursor in all *Conus* peptides [1]. In addition to proteolytic processing, C-terminal amidation, γ-carboxylation of glutamate, hydroxylation of proline, and unusual modifications like epimerization of an L- to a D- amino acid [6], bromination of tryptophan [7], O-glycosylation [8,9,10], cyclization of N-terminal Gln [11], and sulfation of tyrosine [12] have also been observed. Up to 15 post-translationally modified enzymes have been discovered to date, resulting in a total of 10 different types of post-translational modifications in conopeptides (Table 1). 

The post-translationally modified enzymes and secreted products in conopeptides have been reported in a few studies. *Conus* peptidyl proline cis-trans isomerase (PPIase) plays an important role in the folding process of conopeptide and has the ability to induce the cis-trans isomerization of the substrate succinyl-Ala-Ala-Pro-Phe-p-nitroanilide in in vitro enzyme activity assays [13]. Subsequently, it has been demonstrated that protein disulfide isomerase (PDI) cannot only assist the correct formation of disulfide bridges in the presence of a propeptide sequence of conopeptide lt14a but also promote the soluble expression of PPIases and lt14a, raising the possibility that PDI may function as a profusion molecular chaperone [14]. Hydroxylation of proline is also a high-frequency post-translational modification in conopeptides, which can directly affect the folding, structure, and biological activity of conopeptides [15]. In 2018, the sequences of prolyl 4 hydroxylase (P4H) that promoted the hydroxylation of proline were identified by transcriptome sequencing in *Conus literatus* venom [16]. P4H is widely found in a variety of species and is a hetero-4-polymeric α2β2, α subunit contains the peptide-substrate-binding domain and the enzymic active site, and β-subunit is a disulfide isomerase that acts as a helper subunit to promote the soluble expression of α-subunit and ensure its activity in mammalian collagen [17]. However, the hydroxylation machinery of proline in conopeptides is still unclear, which greatly limits the development and application of conopeptides. As an alternative, extract-based cell-free protein synthesis (CFPS) has gained increasing interest to perform the post-translational modification in vivo [18,19,20]. Inspired by the openness, controllability, and convenience of CFPS, we were interested in determining whether P4H might catalyze the hydroxylation of proline through recognition of the special sequence of the prepropeptide precursors in conopeptide. For proof-of-concept studies, μ-conopeptide PIIIA (μ-PIIIA) was selected as a model, which specifically targets Na_V_1.4 with 22 amino acid residues, three disulfide bridges, two hydroxyproline (Hyp) residues, one pyroglutamyl residue, and the amidate of the C-terminus. Our work would provide an innovative synthesis method for multi-modificational conopeptides. 

**Table 1 marinedrugs-21-00421-t001:** Post-translational modification and post-translational enzymes of conopeptides.

Modification	Sites	Post-Translational Enzymes
Disulfide	Cys residues	protein disulfide isomerise, PDI [21]
Hydroxylation	Pro C-4	proline hydroxylase [16] hydroxylase with D-aa specificity [22] lysyl hydroxylase [23]
	Lys C-5	
	Val γ-	
Carboxylation	Glu γ-(vitamin K-dependent)	γ-glutamyl carboxylase [24]
Bromination	Trp C-6	bromoperoxidase [25]
Sulfotyrsine	Tyr	tyrosyl sulfotransferase [12]
Epimerization	Trp, Leu, Phe, and Val	epimerase [26]
Cyclization	Gln	glutaminyl cyclase [11]
O-glycosylation	Ser, Thr	polypeptide hexNAc transferase [8]
Amidation	C terminus	peptidylglycine alpha amidating monooxygenase [27]
Cis-trans isomerization	Pro	peptidylprolyl cis-trans isomerases, PPIases [28]

## 2. Results

### 2.1. A Cell-Free Protein Synthesis Platform for μ-PIIIA Biosynthesis

In this study, we hypothesized that the hydroxylation of proline might depend on the recognition of a special domain in μ-PIIIA [29,30]. Considering the non-splicing peptide precursor in *Escherichia coli*, the enterokinase cleavage site was individually introduced upstream of the signal peptide (pre region), precursor peptide (pro region), and mature peptide (mature region) of µ-PIIIA (Figure 1a) and cloned into pUC57 by Beijing Genomics Institution (BGI, Shenzhen, China) according to the precursor peptide of the µ-PIIIA gene (P01407). The pI-p3a1, pI-p3a2, and pI-p3a3 expression vectors were successfully constructed to generate different hybrid precursors of µ-PIIIA after being digested by Nde I and Xho I and subcloned into the pIVEX-2.4d vector, respectively (Figure 1b). To prove that the continuous exchange cell-free protein system (CE-CFPS) can be used for the synthesis of the precursors of µ-PIIIA, pIVEX-2.4d was performed on the CECF system as a negative control. Western blotting results revealed that all of the hybrid precursors of µ-PIIIA were present in soluble form in 10 μL CECF mixtures, and no target protein was found in the control group (Figure 1c). These results indicated that the precursors of µ-PIIIA could be produced by the CE-CFPS system. 

### 2.2. Use of the CE-CFPS Platform for the Proline Hydroxylation of μ-PIIIA 

The pET-SUMO-P4H (H, AXL97326), pET-SUMO-glutaminyl cyclase (C, ATO8803), and pET-SUMO-PDI (KX494911) vectors that expressed P4H, glutaminyl cyclase, and PDI were reconstructed, respectively (Appendix A), induced by IPTG, and purified by the HisPur^TM^ Ni-NTA affinity chromatography column (Appendix A). P4H and glutaminyl cyclase were presented in soluble form, while they were also highly degradable and sensitive to temperature. As a result, P4H, glutaminyl cyclase, and PDI were employed as plasmids in the CE-CFPS platform, which enables scalable in vitro protein expression with contiguity exchange features to generate exceptional high protein yields. To test the hypothesis, the hybrid precursors of µ-PIIIA in cross combination with PTM enzymes (Table 2) were designed and carried out on the CE-CFPS platform for 24 h. The fusion proteins were purified by the His-tag purification Kit, followed by Enterokinase cleavage at 23 °C for 24 h to remove the His-tag, prepeptide, and propeptide, and the 20 μL purifications were then concentrated to 1 μL by ZIPTIP C18. The concentrated products were then detected by MALDI-TOF MS. Only one combination of pI-p3a3 (containing pro and mature regions of μ-pIIIA) with pET-SUMO-P4H produced recombinant µ-PIIIA (rPIIIA), and the *m/z* (+Na) of rPIIIA was 2632.7 (Figure 2), which was consistent with the theoretical monoisotopic mass (2609.3 Da) of μ-PIIIA with one pyroglutamyl residue and the two hydroxyprolines (Hyps) residues. Considering the absence of peptidylglycine alpha amidating monooxygenase in our study, we presumed that the cyclization of Glu could have instead taken place undergoing the cleavage of Enterokinase and needed our further study. The results suggested that the hydroxylation of proline in μ-PIIIA occurred in the CE-CFPS platform, and P4H might depend on the recognition of the pro region in precursor μ-PIIIA to achieve the hydroxylation of Pro 8 and Pro 18, rather than the mutual assistance of the post-translational modification enzymes. 

### 2.3. Targeted Post-Translational Modification of μ-PIIIA Expression in Escherichia coli

To prove the strategy in CE-CFPS, the vector pET-Trx-p3a3-H for co-expression of P4H and the precursor μ-PIIIA containing pro and mature regions were constructed to express bioactivity peptides in *Escherichia coli*, which has a His-tag for purification and a fusion protein Trx for the correct folding of disulfide bonds in the cytoplasm (Figure 3a and Appendix A). In brief, following the digestion of endonucleases and the sequence verification, the reconstruct was transformed into BL21(DE3) pLysS under IPTG induction for 4 h at 37 °C. Subsequently, the fusion proteins Trx-His_6_-μ-PIIIA were purified by HisPur^TM^ Ni-NTA affinity chromatography columns and eluted by 250 mM imidazole. As shown in Figure 3b, in contrast to the control, target proteins with a molecular weight of ~20 kD were obviously increased, and the yield of fusion proteins was up to 20 mg/L. 

To obtain rPIIIA, the desalted and concentrated fusion proteins were cleavage by Enterokinase at 23 °C for 24 h, and recombinant PIIIA was purified by HPLC. The results showed that rPIIIA was eluted at about 50% solvent B with a retention time of 12.390 (Figure 3c). Subsequently, the molecular mass of rPIIIA was confirmed by MALDI-TOF MS. As predicted, the *m/z* (+H) of rPIIIA was 2604.3 (Figure 3d), which was consistent with the theoretical monoisotopic mass (2603.6 Da) of folded μ-PIIIA with two hydroxyprolines (Hpys) residues and one pyroglutamyl residue and proved the effective and feasible strategy in CE-CFPS. This study marks the first time that the hydroxylation of proline and the cyclization of glutamate in the conopeptide μ-PIIIA are simultaneously achieved on the CE-CFPS platform and in *Escherichia coli*. 

### 2.4. Effect of Recombinant PIIIA on hNav1.4

The biological activity of rPIIIA was evaluated in stable cell lines CHO expressing the human sodium channel Na_V_1.4 (hNav1.4) by whole-cell patch clamp technique. The results showed that rPIIIA could inhibit hNav1.4-mediated ion currents (Figure 4a), and the IC_50_ values for rPIIIA were 939 nM (Figure 4b), which was reduced by almost four times compared to the native μ-PIIIA [31]. Given that the C-terminal amidate of a peptide can influence the bioactivity of conopeptides [27,32], we concluded that this might be related to the absence of an amidate at the C-terminus of recombinant PIIIA. Taken together, rPIIIA with the non-amidate of C-terminus could also antagonize Na_V_1.4. 

## 3. Discussion

CFPS is an open platform based on a cell extract for producing proteins by performing transcription and translation reactions in vitro, such as in *Escherichia coli* [33,34], yeast [35], wheat germ [36], rabbit reticulocyte [37], insect cells [38], Chinese hamster ovarian (CHO) cells [39], and human cells [40], which is supplemented with NTPs, amino acids, creatine phosphate, genetic instructions in the form of DNA, and so on. Compared with traditional in vivo approaches, CFPS has many advantages in protein synthesis. First, it allows convenient monitoring and direct manipulation due to the lack of a cell wall. Second, it saves time because it does not need to break cells to obtain target proteins; plasmid DNA or linear PCR products can be directly added to CFPS reactions. Third, it permits the synthesis of proteins toxic to cells [41,42] and natural products hard to produce in living organisms [43]. Fourth, it does not need to grow, and multiple proteins can be synthesized simultaneously in parallel on microplates, which is especially important for accelerating biological design and high-throughput selection. 

CFPS has emerged as an important approach for accelerating post-translational modification design, which can be built and tested [44]. The N-terminal of the encoding sequence of IgG was fused to an endoplasmic reticulum (ER)-specific signal sequence-melittin peptide, which induced translocation of IgG antibody polypeptide chains into the lumen of ER microsomes, promoted the correct folding of disulfide bridges and the rapid synthesis of antibody molecules in CE-CFPS reactions [45]. Hemagglutinin (HA) stem domain proteins from influenza virus A/California/05/2009 (H1N1) were correctly folded and rapidly produced using *Escherichia coli*-based CFPS [46]. Lanthipeptide Nisin and its analogues were biosynthesised by co-expression vectors of post-translational modification enzymes Nis B and Nis C with Nisin using an *Escherichia coli*-based CFPS platform coupled with a screening assay for anti-gram-negative bacteria growth, and four novel lanthipeptides with antibacterial activity have been identified [47]. CFPS is therefore a suitable tool for the efficient production of peptides with post-translational modification in light of these benefits. As a result, our study aimed to identify P4H with the ability to catalyze the hydroxylation of Pro in μ-PIIIA using CE-CFPS. There have only been a few articles reported on the mechanism of post-translational modification enzymes in conopeptides. Peptidylprolyl cis-trans isomerases (PPIases) not only catalyze the cis-trans isomerization of prolines, but they also increase the rate of the correct folding of the conopeptide μ-GIIIA with three hydroxyprolines in vitro, while having no effect on the folding of the conopeptides ω-MVIIC and μ-SIIIA with no or one proline residue [28]. The catalytic mechanism of γ-glutamyl carboxylase is more obvious than that of other types of post-translational modification in conopeptides. Vitamin K-dependent γ-glutamyl carboxylase has catalyzed the conversion of Glu-Gla through the recognition sequence in the −1 to −20 region of the conantokin-G prepropeptide [48] and in the postpeptide of conopeptides TxX and TxIX [29]. Therefore, we postulated that P4H might catalyze the hydroxylation of proline through the specific recognition or characteristic structure targeted at the precursor peptides of μ-PIIIA, or the mutual assistance of the post-translational modification enzymes [14]. To test this hypothesis, we first reconstructed three hybrid precursor peptides and designed several cross-combinations of hybrid precursor peptides with PTM enzymes (P4H, PDI, and glutaminyl cyclase) to clarify the hydroxylation of proline in μ-PIIIA using the CE-CFPS platform (Table 2). Unexpectedly, the MALDI-TOF/MS results for the combination of pI-p3a3 with P4H showed that the *m/z* (+Na) of rPIIIA corresponded with the theoretical monoisotopic mass of μ-PIIIA with one pyroglutamyl residue and two Hyps residues, indicating that the hydroxylation of Pro 8 and Pro 18 of μ-pIIIA was successfully completed. Considering the absence of peptidylglycine alpha amidating monooxygenase in our study, we presumed that the cyclization of Glu could have instead taken place undergoing the cleavage of Enterokinase and needed our further study. While PDI was unable to facilitate the formation of disulfide bridges in the precursor of μ-pIIIA labeled with histidine. On the contrary, it has been shown that PDI can not only assist in the correct formation of disulfide bridges in the presence of a propeptide sequence of the conopeptide lt14a but can also promote the soluble expression of PPIases and lt14a, suggesting that PDI may also be a kind of profusion molecular chaperone [14]. Our results indicated for the first time that *Conus* P4H might function similarly to γ-glutamyl carboxylase to catalyze the hydroxylation of proline, depending on the recognition of the pro region in the precursor μ-PIIIA. We have learned from the process of hydroxylation that P4H might recognize the pro region in the precursor μ-pIIIA to achieve hydroxyproline and pyroglutamyl using the CECF platform. We next evaluated this strategy in *Escherichia coli*, using the vector pET-Trx-p3a3-H for co-expression of P4H and the precursor μ-pIIIA containing pro and mature regions, which has a His-tag for purification, and a fusion protein Trx for improving the solubility of target peptides and proper folding of disulfide bonds in the cytoplasm. In accordance with the results in CE-CFPS, the hydroxylation of Pro and cylization of Glu occurred simultaneously in the conopeptide rPIIIA, and three pairs of disulfide bonds were also folded properly with the assistance of Trx. This confirmed the effectiveness of the process in CE-CFPS, where P4H catalyzed the hydroxylation of proline through the recognition of the pro region of the precursor μ-PIIIA. Hence, in contrast to the yield of ArIB in our earlier study [49], the production of fusion rPIIIA in *Escherichia coli* was much lower, at approximately 20 mg/L in flasks, which might be attributed to the co-expression of the hybrid precursors of μ-PIIIA and P4H, even though they did not share a common T7 promoter. The post-translational modifications have an important effect on the structure and bioactivity of conopeptides. Subsequently, the pharmacological properties of rPIIIA were evaluated by electrophysiology. rPIIIA exerted an inhibitory effect on the ion current of hNa_V_1.4 with IC_50_ values of 939 nM, and the activity of rPIIIA was 4-fold less active in inhibiting hNa_V_1.4 than native μ-PIIIA [31]. By reason of the foregoing, we speculated that the aminate of the C-terminus of rPIIIA biosynthesized in our study might not form, which is directly related to the stability and bioactivity of conopeptides [32]. In conclusion, the study first established that P4H catalyzed the hydroxylation of proline through recognition in the pro region of the prepropeptide precursor in μ-PIIIA, and rebuilt an efficient and feasible method to produce the bioactive μ-PIIIA with multi-posttranslational modifications using CE-CFPS as the synthesis platform. This finding may offer an alternative insight into synthesizing multiple post-translationally modified conopeptides.

## 4. Materials and Methods

### 4.1. Strains and Agents

The *Escherichia coli* strains DH5α and BL21(DE3) pLysS, as well as the expression vector pET-32a containing the Trx tag, were purchased from Novagen (Darmstadt, Germany); pIVEX 2.4d was purchased from Roche (Basel, Switzerland); and pET-SUMO-28a was constructed and conserved in our lab. The RTS^TM^ 100 *Escherichia coli* Disulfate Kit was purchased from Biotechrabbit GmbH (Berlin, Germany). Restriction enzymes Kpn I, EcoR I, Not I, and Mlu I, T4 DNA ligase, protein markers, protein HisPur^TM^ Ni-NTA Purification Kit, HAM’S/F12 medium, Fetal Bovine serum (FBS), penicillin, and streptomycin were purchased from Thermo Scientific (Waltham, MA, USA). Enterokinase was purchased from New England Biolabs, Inc. (Ipswich, MA, USA). Vydac C18 218TP54 columns (5 μm, 4.6 mm × 250 mm, 10 μm, 22 mm× 250 mm) were purchased from Grace (Deerfield, IL, USA). Acetonitrile (ACN, gradient grade for HPLC), and other chemical reagents were all of analytical grade and purchased from Sigma-Aldrich (St. Louis, MO, USA).

### 4.2. Cell Culture

Stable cell lines expressing CHO and hNa_V_1.4 were constructed and conserved in our lab until use. Cells were grown in monolayers and cultured in HAM’S/F12 medium supplemented with 1% (*v/v*) penicillin-streptomycin (100 μg/mL penicillin, 100 μg/mL streptomycin) and 10% (*v/v*) FBS. Cells were grown in 25 cm^2^ tissue culture flasks at 37 °C in an atmosphere of 5% CO_2_ in a humidified incubator and passaged every 2 3 days.

### 4.3. Plasmid Construction

For the expression of μ-PIIIA in CFPS, several plasmids were constructed. The plasmids pUC57-p3a1 containing μ-PIIIA pre, pro, and post domains, pUC57-p3a2 containing μ-PIIIA pre and post domains, and pUC57-p3a3 containing μ-PIIIA pro and post domains, pUC57-prolyl 4-hydroxylase (pUC57-H), pUC57-Glutamyl carboxylase (pUC57-C), and pUC57-PDI (pUC57-PDI) were individually synthesized by Nanjing Genscript Biological Technology Limited Company (Nanjing, China). The fragments containing p3a1/p3a2/p3a3/prolyl 4-hydroxylase/Glutamyl carboxylase/PDI digested by Nde I and Kpn I were individually subcloned into the corresponding sites of the plasmid pIVEX-2.4d (Novagen, Darmstadt, GER). All fragments obtained by digestion were gel-purified using a DNA gel extraction kit (Axygen, Corning, NY, USA) according to the manufacturer’s instructions before cloning.

For the biosynthesis of prolyl 4-hydroxylase (H), Glutamyl carboxylase (C), PDI, and μ-PIIIA in *Escherichia coli*, expression plasmids pET-SUMO-H, pET-SUMO-C, pET-SUMO-PDI, and co-expression plasmid pET-Trx-p3a3-H were constructed individually. The primers used for plasmid construction are listed in Table 3. In general, the primers named after the plasmid were used to amplify and construct the relevant plasmids, and the corresponding restriction enzyme sites were designed on the primers. For instance, pET-PIIIA-F and pET-PIIIA-R were used for p3a3 amplification, and the PCR fragment was cloned into the EcoR I and Sac I sites of pET-Trx and yielded pET-p3a3, which was constructed to express the p3a3 gene with a His6 tag. In the plasmid pET-p3a3-H with a T7 promotor, the P4H gene was inserted between the Xba I and Not I sites of the plasmid pET-p3a3. 

### 4.4. CE-CFPS Reactions

CE-CFPS reactions were performed to synthesize μ-PIIIAs. Briefly, the RTS^TM^ 100 *Escherichia coli* Disulfate Kit supplemented with essential components was used for in vitro transcription and translation of μ-PIIIA PTMs. Reactions were mixed into 50 μL according to Table 2. For each reaction, plasmids were added in various combinations in a total quantity of 0.5 μg (Appendix A). The CE-CFPS reactions were performed at 900 rpm at 32 °C for 24 h. Reactions were terminated by placing on ice, and precipitated proteins were removed by centrifugation at 10,000× *g* for 10 min. The resulting supernatant was subjected to the next analysis.

### 4.5. Purification and Identification of rPIIIA 3

The protein rPIIIA 3 was purified according to the following protocol: In brief, the recombinant plasmid pET-p3a3-H was transformed into strain BL21 (DE3) of Escherichia coli to express the fusion protein His_6_-PIIIA. A single colony was picked and cultured in 10 mL of 2 × YT medium supplemented with 50 μg/mL Kan at 37 °C overnight. Then they were inoculated into 1 L of 2 × YT medium at a ratio of 1:100 and grown at 37 °C until the OD_600_ reached 0.6–0.8 and were induced with 0.1 mM isopropyl-β-D-1-thiogalactopyranoside (IPTG) for 4 h at 37 °C. The cells were harvested by centrifugation at 4500 rpm for 15 min at 4 °C, resuspended in 100 mL of ice-cold lysis buffer (20 mM PBS, pH 7.4), lysed by homogenization using a low temperature ultra-high pressure continuous flow cell disrupter, and then centrifuged at 12,000× *g* for 30 min at 4 °C. After being centrifuged and filtered through the 0.45 μm filter, the supernatant with His_6_-PIIIA 3 was purified by the Ni-NTA column (GE Healthcare, Marlborough, MA, USA). Then the recombinant fusion protein of His_6_-PIIIA 3 was eluted using imidazole at three incremental concentrations of 125, 250, and 500 mM. All of the eluted fractions were collected and analyzed by 15% SDS-PAGE gel. The purified His_6_-PIIIA 3 solution was desalted, concentrated with an ultrafiltration spin column (NMWL 3kDa, Millipore, Billerica, MA, USA) at 12,000× *g* for 30 min, lyophilized and stored at −20 °C until used.

The protein rPIIIA was prepared according to our previously described method [47]. In brief, the ratio of the fusion protein and Enterokinase was 1:0.0006% (M:M) in Enterokinase digestion buffer (100 mM NaCl, 4 mM CaCl_2_, 40 mM Tris·HCl, pH 8.0) at 23 °C for 24 h. The supernatant containing the rPIIIA peptide cleaved by Enterokinase was applied to HPLC (Agilent 1100, Agilent Technologies Inc, Los Angeles, CA, USA) with a Vydac C18 semi-preparative column (10 μm, 22 mm × 250 mm). The peptide elution was performed with a linear gradient of 0–50% solvent B over 25 min at a flow rate of 4 mL/min, where solvent B was 100% ACN with 0.1% (*v/v*) TFA; solvent A was 0.1% (*v/v*) TFA. Absorbance was monitored at 214 nm.

### 4.6. Qualitative Analyses by MALDI-TOF MS

Qualitative analyses of rPIIIA and other analogues were performed using a previously described MALDI-TOF/MS-based method with modifications [47]. In summary, after centrifugation at 10,000× *g* for 10 min, the supernatant of the CFPS reaction mixture or other purified solutions treated with Enterokinase were desalted using ZIPTIP C18 (Millipore, Billerica, MA, USA) and subjected to MALDI-TOF MS. MALDI-TOF MS analysis was performed using an Ultraflex MALDI-TOF mass spectrometer (Bruker Daltonics, Billerica, MA, USA), equipped with a 50 Hz pulsed nitrogen laser (λ = 355 nm) and a 19 KV accelerating voltage operated in reflectron, positive ion mode. The samples were prepared by mixing 1 μL peptide solution with 1 μL of saturated matrix (α-cyano-4-hydroxycinnamic acid) in 0.1% TFA containing 30% ACN. Data collection and processing were respectively performed by FlexControl and FlexAnalysis software (V2.4, Bruker Daltonics, Billerica, MA, USA).

### 4.7. Western-Blot Assay

The CE-CFPS mixture was prepared according to the protocol. Briefly, the proteins from the supernatants of CE-CFPS were separated by SDS-polyacrylamide gel electrophoresis and transferred to polyvinylidene fluoride membranes (Millipore, Billerica, MA). The membranes were probed with polyclonal rabbit antibodies against anti-His (Abcam, Cambridge, UK) overnight at 4 °C and incubated for 1 h with IRDye 800 conjugated with affinity purified anti-Rabbit IgG antibody (Rockland, Philadelphia, PA, USA). The membranes were scanned by the Odyssey Infrared Imaging System (LI-COR Bioscience, Lincoln, NE, USA). CE-CFPS was performed at 32 °C for 24 h.

### 4.8. Whole Cell Patch-Clamp Recordings

The currents of cells were recorded according to a previously described method [50]. In brief, whole-cell patch-clamp recordings were performed in an extracellular solution containing 140 mM NaCl, 3.5 mM KCl, 1 mM MgCl_2_, 2 mM CaCl_2_, 10 mM D-Glucose, 1.25 mM NaH_2_PO_4_, and 10 mM HEPES. Solutions were adjusted to pH 7.4 with NaOH. Pipettes were filled with intracellular solutions containing 50 mM CsCl, 10 mM NaCl, 10 mM HEPES, 60 mM CsF, 20 mM EGTA, CsOH, and 10 HEPES. All recordings were made using an EPC-10 patch-clamp amplifier (HEKA Elektronik, Reutlingen, Pfalz, GER). The resistance was <5 MΩ for all recordings. Multiple concentrations were detected for each test compound. Before each protocol, the cell was voltage-clamped at a holding potential of −130 mV to ensure complete removal of both fast and slow inactivation. All recordings were conducted at room temperature. Cells were initiated to be treated with a dose of drug for 5 min until whole-cell recorded Na_V_1.4 current stabilization, and then the next concentration was detected; each cell was detected at multiple concentrations. Measurements from 3 to 6 cells were conducted in each treatment group, and all data were presented as the mean ± S.D. Estimates of potency were obtained by fitting concentration response curves to the data by the equation: Inhibition = 1/[1 +(IC_50_/C_toxin_) n] with non-linear regression analysis using IGOR Pro 7 (IGOR Software, V7.0.8.1, WaveMetrics, Inc., Oswego, NY, USA), where n is the Hill coefficient and IC_50_ is the antagonist concentration giving a half-maximal response [rPIIIA].

## Figures and Tables

**Figure 1 marinedrugs-21-00421-f001:**
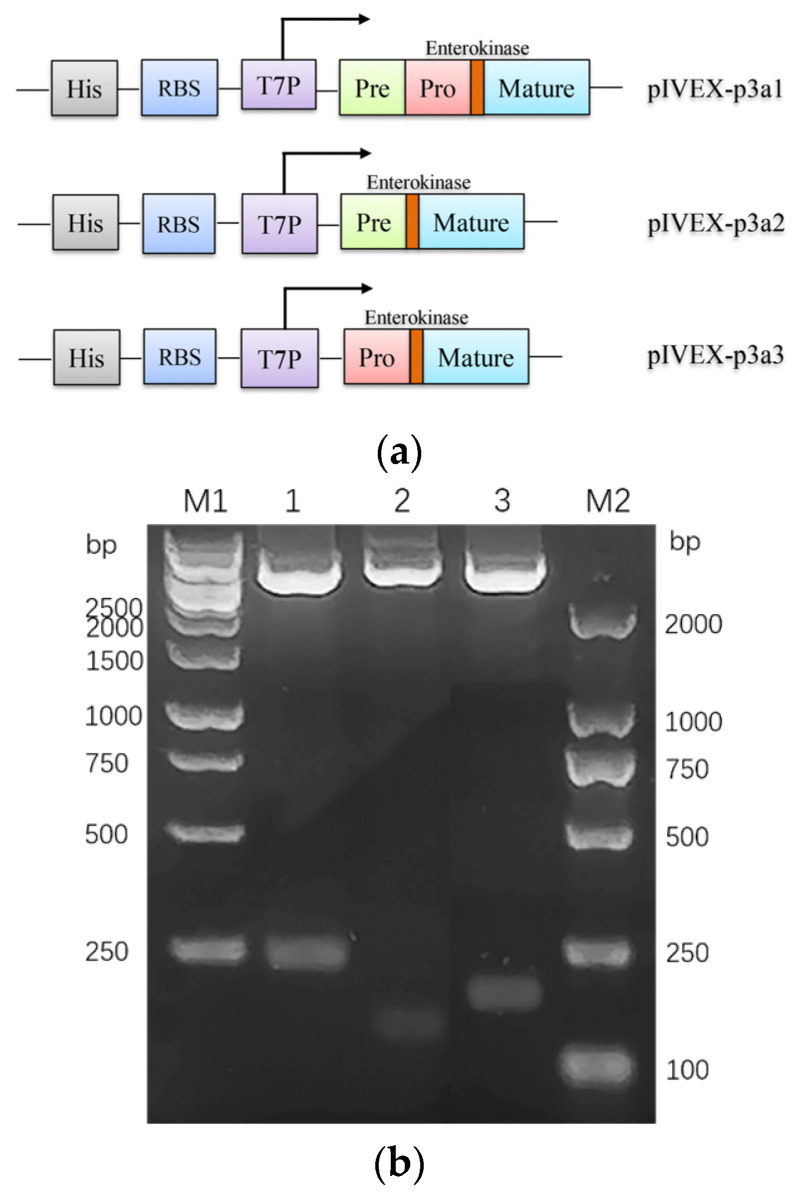
Reconstitution of the precursors of µ-PIIIA biosynthesis using CECF platform. (**a**) Schematic illustration of pI-p3a1, pI-p3a2, and pI-p3a3 reconstitution. The synthetic genes coding for p3a1, pI-p3a2, and pI-p3a3 were inserted downstream of His6 tag between Nde I and Xho I sites of the expression vector pIVEX-2.4d, respectively, and an enterokinase site was introduced at the upstream of the pre, pro, and post regions of µ-PIIIA for cleavaging the His6-tag, prepeptide, and propeptide. (**b**) The gel image of plasmids pI-p3a1, pI-p3a2, and pI-p3a3 digested by Nde I and Xho I, respectively; M1, 1 kb DNA Ladder; M2, DL2000 DNA Ladder. (**c**) Western-blotting analysis of p3a1, p3a2, and p3a3 expressed in CE-CFPS.

**Figure 2 marinedrugs-21-00421-f002:**
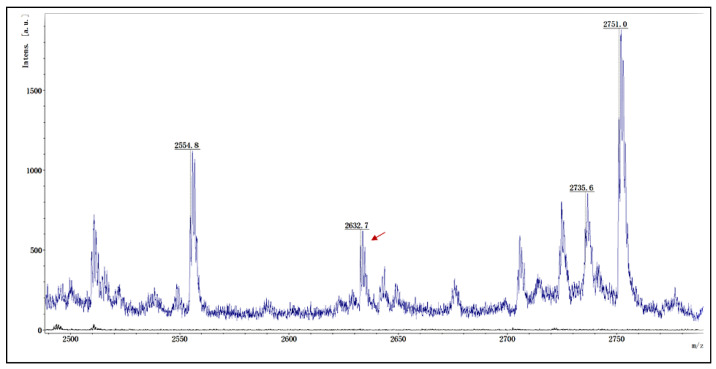
MALDI-TOF MS analysis of purified rPIIIA from CE-CFPS. The *m/z* (+Na) of rPIIIA with observed molecular weight of 2632.7 Da, which was consistent with calculated theoretical monoisotopic mass of 2609.3.

**Figure 3 marinedrugs-21-00421-f003:**
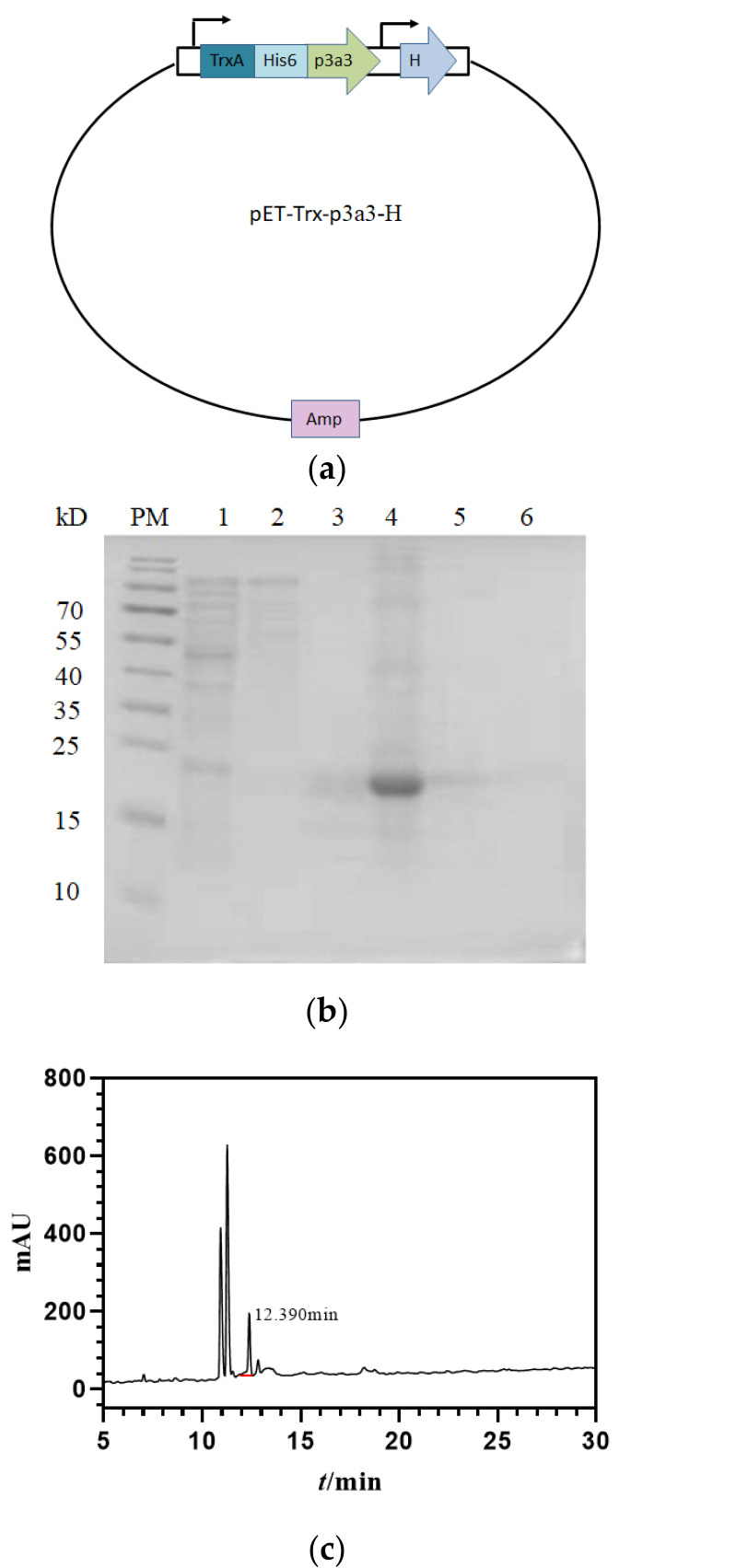
CE-CFPS platform guided the overproduction of recombinant µ-pIIIA. (**a**) Schematic illustration of pET-p3a3-H co-expression vector. (**b**) 15% SDS-PAGE analysis of recombinant pET-PIIIA3 purified by HisPur^TM^ Ni-NTA affinity chromatography column. PM: multicolor low-range protein ladder (Thermo Scientific 26616). Lane 1: uninduced recombinant pET-PIIIA3; Lane 2: washed fractions using 25 mM imidazole; Lane 3~4: eluted fractions using 125 mM and 250 mM imidazole, individually; Lane 5~6: eluted fractions using 500 mM imidazole. (**c**) RP-HPLC chromatogram of purified rPIIIA. rPIIIA following enterokinase cleavage of thioredoxin was analyzed by RP-HPLC on a Vydac C18 column (5 μm, 4.6 mm × 250 mm), using a linear gradient of 0–50% solvent B over 25 min at a flow rate of 1 mL/min. Solvent B was 100% ACN and 0.1% (*v/v*) trifluoroacetic acid (TFA), and solvent A was 0.1% (*v/v*) trifluoroacetic acid (TFA). Absorbance was monitored at 214 nm. (**d**) MALDI-TOF MS analysis of purified rPIIIA. The *m/z* (+H) of rPIIIA with observed molecular weight of 2604.3 Da was consistent with calculated theoretical monoisotopic mass of 2603.6.

**Figure 4 marinedrugs-21-00421-f004:**
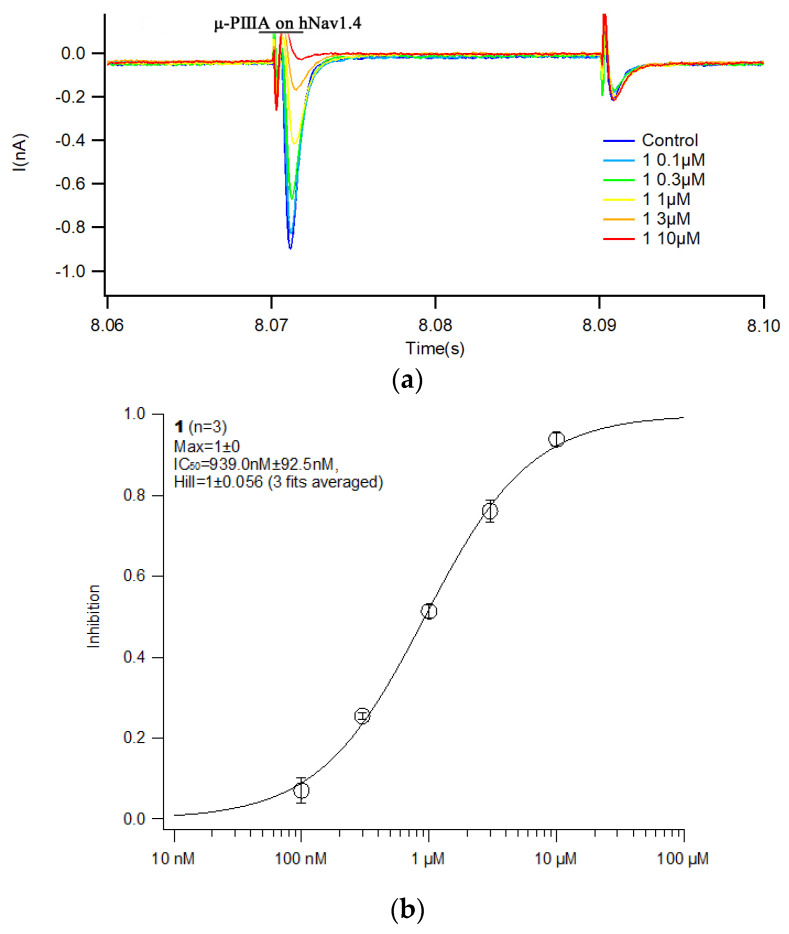
Analysis of Nav1.4-CHO current block by rPIIIA. (**a**) rPIIIA blocks the currents of hNav1.4 in doses of 0.1 μM, 0.3 μM, 1 μM, 3 μM, and 10 μM; (**b**) rPIIIA blocks hNav1.4 with an IC_50_ = 939 ± 92.5 nM, nH = 1 ± 0.056. h, human; nH, Hill slope.

**Table 2 marinedrugs-21-00421-t002:** Cross-combination of hybrid precursor peptides with PTM enzymes.

	pI-p3a1-2.4d			pI-p3a2-2.4d			pI-p3a3-2.4d		
pH-2.4d	+	+	+	+	+	+	+	+	+
pC-2.4d	+			+			+		
pPDI-2.4d	+	+		+	+		+	+	

The total quantity of pI-p3a1-2.4d, pI-p3a2-2.4d, and pI-p3a3-2.4d was 0.4 μg, and the total quantity of pH-2.4d, pC-2.4d, and pPDI-2.4d was 0.1 μg in cross-combinations of CE-CFPS.

**Table 3 marinedrugs-21-00421-t003:** The primer sequence used in this study.

Name	Sequence (5′–3′)
pET-PIIIA-F	cgGAATTCATGCTGCCGATGGACGGTGA
pET-PIIIA-R	GAGCTCTTAGCAGCAACGGTGCGGT
pET-H-F	gcCTCGAGATGAAACTGACCGGTCCGGCGC
pET-H-R	GCGGCCGCTTACTGAATAACCTCTTTCAGCG
pET-SUMO-PDI-F	ggGGTACCAAGTTCAGCAGCTGCCTG
pET-SUMO-PDI-R	CTCGAGTTACAGTTCATCACGCGG
pET-SUMO-H-F	ggGGTACCAAACTGACCTGGACCACCAC
pET-SUMO-H-R	CTCGAGTTACTCAATGATTTGCTTGATCGC
pET-SUMO-C-F	ggGGTACCATGGAGAAGGTGACCACCG
pET-SUMO-C-R	CTCGAGTTACAGCAGATCAACCAGAAACAG

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
