# Peer review of "Usage of Cell-Free Protein Synthesis in Post-Translational Modification of μ-Conopeptide PIIIA"

_marinedrugs, 2023, doi:10.3390/md21080421_

Round 1
Reviewer 1 Report
The MS might be an interesting contribution, but I consider that a report right now might not be fair because the intention of several important parts are really difficult to fully understand due to the English.
Unfortunately, I found that English is still pretty inappropriate In the revised (for English) version. Even though some issues of the English are just a matter of style, the MS contains too many serious grammatical errors that make understanding some sections difficult and time consuming. I consider that a fair evaluation cannot be done with the current version.
Author Response
Dear reviewer:
Thanks for your hard and serious review! We accepted your all suggestions, and have reedit the English and seriously revised. Please see the manuscript-revised.
Reviewer 2 Report
The manuscript described the production of conopeptide with post-translational modifications using a cell-free protein production system and an in vivo bacterial expression system.
I have some comments before considering the manuscript
1) All the scientific names (+in vivo) should be in italics
2) Line 81: “…multi-modification conopeptides” should read as “…multi-modification of conopeptides.”
3) change “pro” to “Pro”
4) Figure 1 is labeled Figure 2.
5) “…signal peptide (pre region), precursor peptide (pro region), or mature peptide 100
(post region)…”. What is this at the end of the paragraph on Page 3?
6) Figure 1C: what are the bands at higher molecular weight regions? Any aggregates of the peptide?
7) use one type of abbreviation for cell-free protein synthesis.
8) the labels in Fig. 3f are not readable.
9) are the low-intensity peaks in the mass-spec (Fig. 3f) from non-modified peptide? Explain.
10) Figures need to be organized. Some of the DNA gel images can be moved to SI.
11) In section 2.2, the observed mass is ~2632 Da, and the theoretical mass is ~2609 Da. The values are very different. Explain.
The manuscript is hard to follow; the text needs major changes in presenting the data throughout.
Reviewer 3 Report
This article is devoted to the usage of cell free protein synthesis in post-translational modification of µ-conopeptide Pâ…¢A. The authors performed expressions and analyzed the data obtained. This work is very interesting and may be useful to many researchers who deal with small active peptides. However, the careless design of the manuscript and a large number of spelling errors greatly complicate the understanding of the material. The manuscript needs to be carefully revised.
Line 14, 121: “was” needs to be replaced with “were”
Lines 32-34: The phrase “Conopeptides are a kind of small bioactive peptides ribosomally synthesized in-cluding multiple disulfides and a large number of post-translational modifications (PTMs), which is of the structural stability, target specificity, relatively small size and regarded as a rich source of molecular probes in neuroscience” should be rephrase for better understand.
Line 43: “stabilize” needs to be replaced with “stabilizes”
Line 44: conatokins? What do you mean?
Line 45: “is” needs to be replaced with “are”
Line 51: “Sulfatation” needs to be replaced with “sulfatation”
Line 66: “C. amadis” must be written in full and in italics; the end of the phrase “…. sequencing in conopeptides C. amadis venom” should be rephrase for better understand.
Line 75: “in vivo”? What do you mean?
Lines 63-65 and 75-76: should be rephrase, repeat with abstract
Line 86: “pro” needs to be replaced with “Pro”
Line 93: “vector” needs to be replaced with “vectors” and what do you mean “respectively”?? The phrase “Then subcloned into the pIVEX-2.4d vector digested by Nde I and Xho I, the pI-p3a1, pI-p3a2, and pI-p3a3 expression vector were successfully constructed, respectively (Figure 1 b).” should be rephrase for better understand. Figure 1 b does not demonstrate construction creation, it is a gel image of digested plasmids.
Line 93: “gene” needs to be replaced with “genes”
Line 108: “PL2000” - what is it?
Line 112: The phrase “vector expressed Proline hydroxylase, glutaminy l cyclase and PDI” needs to be replaced with “vectors expressed proline hydroxylase, glutaminyl cyclase and PDI, respectively”
Lines 114-116: The phrases “The P4H and glutaminyl cyclase were pre-sented in soluble form, while sensitive to temperature, and particularly unstable and easily degraded. Therefore, P4H, glutaminy l cyclase and PDI were used in form of plasmids in CECF system” should be rephrase for better understand. One does not follow from the other. The output is incorrect.
Line 120: “to remove His-tag”, Why only His-tag?
All experimental details need to be moved to M&M (lines 120-121 and 145-149)
Line 122: “purifications” needs to be replaced with other termin.
Line 124: The accuracy of the MALDI method does not allow mass to be estimated with such accuracy. Leave only an integer in the text.
Line 126: “site” needs to be replaced with “sites”
Lines 143, 259: “domain” needs to be replaced with “domains”
Lines 158-159: It is not clear how these masses are consistent with the description of modifications and masses in Section 2.2. Please match. It is not clear how cyclization is proven.
Lines 201, 202, may be “they” instead “it”?
Lines 239-242: Should be removed all experimental details, please, discuss don't list!
Lines 258, 270: “E. coli” needs to be wrote
After correcting the grammar, substantive questions may appear. At this stage it is very difficult to read. There are also many errors in the methods. I strongly recommend the authors to edit the English language.
I strongly recommend the authors to edit the English language.
Round 2
Reviewer 1 Report
The MS is an interesting contribution, but this reviewer considers that still there are several important parts that are difficult to fully understand due they are not adequately described (for example, those related to the plasmids encoding the enzymes that carry out the PTMs: PDI, glutaminyl cyclase, and prolyl hydroxylase).
The English is still inappropriate in the revised version; this problem needs to be fixed.
Reviewer 2 Report
The revised manuscript reads better. However, some of the previous comments still apply to the revised version of the manuscript.
1) Line 81: “…multi-modification conopeptides” should read as “…multi-modification of conopeptides.”
2) All the organism names in the main text and references should be in italics.
3) "...in vivo/in vitro..." should be in italics.
4) Some of the text is repeated. For example...
"All of the eluted fractions were gathered and subjected to SDS-PAGE analysis. All of the eluted fractions were collected and analyzed by SDS-PAGE."
5) The manuscript needs careful reading.
T
Reviewer 3 Report
Accept in present form
Accept in present form
Author Response
Dear reviewer:
Thanks for your hard and serious review! We have been extensive English and seriously revised. Please see the revised manuscript.